# The Frequency of Daily Consumption of Sugar-Sweetened Beverages Is Associated with Reduced Muscle Mass Index in Adolescents

**DOI:** 10.3390/nu14224917

**Published:** 2022-11-20

**Authors:** Maylla Luanna Barbosa Martins Bragança, Carla Cristine Nascimento da Silva Coelho, Bianca Rodrigues de Oliveira, Eduarda Gomes Bogea, Susana Cararo Confortin, Antônio Augusto Moura da Silva

**Affiliations:** Public Health Department, Federal University of Maranhão, São Luís 65020-070, Maranhao, Brazil

**Keywords:** beverage, sugar, lean mass, adolescents, observational study

## Abstract

The consumption of sugar-sweetened beverages (SSBs) has increased in recent years and has become a cause of concern because these beverages pose a risk to human health. Thus, we evaluated the association between SSBs consumption and muscle mass index (MMI) in adolescents. This cross-sectional study evaluated 2393 adolescents (18/19-years-old). Consumption of SSBs was analyzed based on the frequency of daily consumption and energy contribution categorized into tertiles. MMI was examined using the ratio of muscle mass (kilograms) to height (meters squared). The highest tertile of the frequency of daily SSB intake was associated with a reduced MMI in men (β = −0.31; 95%CI: −0.60, −0.01) and women (β = −0.24; 95%CI: −0.45, −0.02). However, these associations were not observed after adjusting for sugar contained in SSBs in men (β = −0.26; 95%CI: −0.69, 0.17) and for carbohydrate, lipid, and protein intake in women (β = −0.19; 95%CI: −0.42, 0.04). The highest energy contribution tertile of SSBs was associated with a reduced MMI in male adolescents (β = −0.34; 95%CI: −0.64, −0.04). This association was not observed after adjusting for intake of sugar in SSBs (β = −0.38; 95%CI: −0.75, 0.01). The frequency of daily consumption of SSBs was considered a risk factor for decreased MMI in both sexes, and the energy contribution of these drinks was a risk factor for MMI reduced only in male adolescents.

## 1. Introduction

The consumption of sugar-sweetened beverages (SSBs) has increased in recent years worldwide [1]. Moreover, adolescents have an increased intake of SSBs because they are sweet, ready for consumption, highly palatable, and can be easily accessed [2]. In addition, young individuals are more likely to have unhealthy eating practices, reducing milk and water intake and increasing SSBs consumption [3].

The consumption of SSBs has become a cause of concern because these beverages pose a risk to human health, including increased mortality [4]. The consumption of SSBs implies increased body fat because it is the main source of added sugar in foods and because sugar affects satiety lesser than other isoenergetic foods [5].

Few studies have investigated the associations between food intake and muscle mass in adolescents and young adults. In fact, we only found one study that analyzed this association among individuals aged 14–18 years from Georgia, United States [6]. In that study, the consumption of SSBs was negatively associated with muscle mass. The sugars in these drinks interfere with muscle cell autophagy [7], telomere shortening, cell senescence, and apoptosis [8]. Thus, this mechanism may be correlated to muscle cell functionality and muscle mass reduction.

The association between SSB consumption and muscle mass must be investigated, as some studies have observed that loss of muscle mass may predict cardiovascular diseases and life-long mortality [9,10,11]. In addition, as progressive loss of muscle mass occurs in adults and middle-aged individuals and the rate of loss is accelerated and maintained during aging, factors that can lead to early loss of muscle mass must be assessed [12]. These findings may help alleviate reduced muscle mass in adults and older individuals [7,12].

Considering the increase in SSB consumption among adolescents [13] and the possible deleterious effect of consuming these beverages on muscle mass, this study aimed to analyze the association between the consumption of SSBs and muscle mass index (MMI) in adolescents.

## 2. Methods

### 2.1. Study Design

This cross-sectional analysis used data from the São Luís Birth Cohort study, 1997/1998, which is part of the Brazilian Birth Cohort Consortium. The birth cohort at baseline included babies born to mothers who gave birth at hospitals in the municipality of São Luís from March 1997 to February 1998. The study was conducted in 10 public and private hospitals. Systematic sampling with stratification that is proportional to the number of births in each hospital was performed. In this phase, 2542 babies participated in the study. After excluding stillbirths, the sample of this phase included a total of 2443 live births [14]. At 18 and 19 years, the participants in this cohort underwent a new assessment from January 2016 to December 2016. To locate the participants, searches were performed in schools, universities, addresses, and contacts. A total of 654 adolescents participated in this stage of the study. Due to difficulties in locating individuals and increasing the size of the study sample, other adolescents born in the city of São Luís in 1997 were also included. The new members were selected by random sampling from the System of Information about Liveborns restricting for children born in 1997. The new participants were identified via school or university registration and military enlistment registration. Thus, 1861 adolescents were included in the survey. This phase included 2515 adolescents. The data of 2393 participants were considered in this study. However, 122 participants had missing information about muscle mass (Figure 1). A sample of 2393 adolescents had a statistical power of 76.59% to detect a difference of 2.86 kg/m^2^ in MMI mean between SSB consumption groups, with a probability of type I error of 5%.

For such, the individuals were included in two ways: through a raffle using the Live Birth Information System (SINASC in the Portuguese acronym) and also as volunteers identified in schools and universities.

### 2.2. Data Collection

Trained health professionals performed data collection. The information was recorded in the Research Electronic Data Capture (Redcap^®^) online program [15], using standardized questionnaires.

### 2.3. Independent Variables

The frequency of daily consumption and the energy contribution of SSBs categorized in tertiles were considered as exposure variables. SSB consumption was evaluated using a food frequency questionnaire (FFQ). This instrument was developed by Schneider et al. [16] and was validated by Bogea et al. [17]. Considering the last 12 months, eight options of answers were considered to evaluate the frequency of consumption of each food item: never or <1 time/month; 1–3 times/week; 2–4 times/week; 5–6 times/week; 1 time/day; 2–4 times/day; ≥5 times/day. To estimate average portion sizes, a database with pictures of portion sizes of each food was showed to each participant. The adolescent was questioned if they consumed the average portion size, a bigger amount (1.5 of the average portion), or a smaller amount (0.5 of the average portion). The frequency of consumption of each food item and portion sizes were converted into annual consumption and, subsequently, into daily consumption. Food intake in grams was converted into consumed energy (kcal/day) using Food Composition Tables [18,19,20] and food labels. The SSBs included in this study were soft drinks, industrialized juices (fruit juice, pulp or plant extract with the addition of sugars, colorings, and flavorings), and chocolate drinks because they are industrially sweetened and contain the highest amount of added sugars [21].

### 2.4. Dependent Variable

The outcome variable was muscle mass assessed using the MMI calculated by muscle mass (kilograms) divided by height (meters squared). We chose to analyze MMI to correct for height and consider the distribution of muscle mass in the body [22]. In relation to this, the height (in centimeters) was obtained using a stadiometer (Altura Exata^®^, Minas Gerais, Brazil). Total body mass was calculated in kilograms using the Filizola^®^ scale (Sao Paulo, Brazil), and muscle mass (kilograms) was assessed using an enCORE-based dual-energy X-ray densitometry (DXA) and the Lunar Prodigy (GE Healthcare Systems). Due to the differences observed between total body weight estimated using DXA and the total body weight of the Filizola brand scale, DXA muscle mass adjustment was performed. Initially, the percentage of muscle mass was calculated based on the total body weight estimated using DXA. The percentage of muscle mass was considered for the calculation of the adjusted muscle mass (kilograms) using the total body weight of the scale [23].

### 2.5. Complementary Variables

The confounding factors were as follows: sex (female and male), age (18–19 years), self-declared skin color (white, black, and brown) [24], education (elementary, high school, technical or vocational course, and college), economic class according to the criteria of the Brazilian Association of Research Companies (ABEP; A, B, C, and D–E) [25], physical activity assessed using the Self-Administered Physical Activity Checklist [26], current consumption of alcoholic beverages (no and yes), current smoking status (no and yes), sleep duration (<8 and ≥8 h) [27], and consumption of carbohydrates (grams/day), lipids (grams/day), and proteins (grams per day/kg body weight) as well as sugars (grams/day) in SSBs.

### 2.6. Data Analysis

Data were analyzed using the STATA^®^ statistical software version 14.0. The chi-square test was used to assess differences in terms of sex and categorical variables, and Student’s t-test was utilized to examine differences in terms of sex and continuous variables. To validate the association between SSB consumption and MMI, linear regression was performed as follows: Model 1: unadjusted analysis; Model 2: adjusted for age, skin color, education, and socioeconomic status; Model 3: adjusted for model 2 variables as well as physical activity, smoking status, alcohol consumption, and sleep duration; Model 4: adjusted for model 3 variables as well as carbohydrate, lipid, and protein intake; Model 5: adjusted for model 4 variables as well as intake of sugar in SSBs. In model 5, BAA sugars were studied as mediators in case their adjustment in the analysis made the association between BAA and MMI disappear [28]. The 95% confidence interval (95% CI) without the inclusion of zero was considered significant. To evaluate the effect size of the associations assessed, a standardized coefficient (standardized β) was used, which indicates the change in a certain value of the standard deviation in the outcome variable caused by the change in the standard deviation in the exposure variable. A small effect size was considered when the value was approximately 0.10 [29]. Differences were found in the associations between the consumption of SSBs and MMI in terms of sex. Thus, the analyses were conducted independently.

### 2.7. Ethical Considerations

The study was approved by the Research Ethics Committee of University Hospital of the Federal University of Maranhao (n°:1.302.489). All participants signed an informed consent form.

## 3. Results

A total of 2393 adolescents were evaluated. Among them, 52.6% were women. Most male and female adolescents were aged 18 years (70.9% and 66.1%), had brown skin color (65.6% and 62.7%) and high school education (69.6% and 63.7%), belonged to economy class B (51.5% and 46.1%), slept less than 8 h a day (55.5% and 55.9%), and did not smoke (95.1% and 97.9%) and consume alcoholic beverages (52.9% and 62.7%, respectively). Among the male adolescents, 58.8% did not engage in sufficient physical activities. Meanwhile, among the female adolescents, 80.8% were physically active (Table 1).

The mean MMI values of the male and female participants were 16.3 and 13.3 kg/m^2^, respectively. The mean SSB consumption was 0.9 times a day in both sexes, accounting for 2.9% and 2.7% in relation to the total calories in the diet, with daily averages of 2.1 and 2.0 times for the highest tertile frequency of the consumption of SSBs and values of 6.2% and 6.7% for the SSB’s largest energy contribution tertile. With regard to sex, the male adolescents had higher mean MMI values (16.3 kg/m^2^) and a higher consumption of carbohydrate (504.9 g/day) and lipids (83.9 g/day) than the female adolescents (Table 2).

In the male adolescents, the third tertile of the frequency of SSB consumption (daily consumption between 1.1 and 10.1 times/day) was associated with a reduction in MMI by 0.31 kg/m^2^ (95%CI: −0.60, −0.01; standardized β = −0.070). In female adolescents, the increase in the third tertile of SSB consumption frequency was associated with a reduction in MMI by 0.24 kg/m^2^ (95%CI: −0.45, −0.02; standardized β = −0.068). These associations were not observed after adjusting for the intake of sugar in SSBs in men (β =−0.26; 95%CI: −0.69, 0.17; standardized β = −0.060) and for the intake of carbohydrate, lipid, and protein in women (β = −0.19; 95%CI: −0.42, 0.04; standardized β = −0.051) (Table 3).

Each increase in the third tertile of the SSB energy contribution (with values ranging from 3.0% to 28.4% in relation to total calories) was found to be associated with a reduction in MMI by 0.34 kg/m^2^ in male adolescents (95% CI: −0.64,−0.04; standardized β = −0.080). This association was not observed after adjusting for intake of sugar in SSBs (β = −0.38; 95% CI: −0.75; 0.01; standardized β = −0.076) (Table 4).

## 4. Discussion

The main results of this study showed that the higher frequency of SSB consumption was associated with a reduced MMI in both sexes and the highest energy contribution of SSBs was associated with a reduced MMI in male adolescents. The intake of sugar in SSBs and macronutrients of food may be the mechanism underlying the association in men and women, respectively.

The average frequency of 2.0 times a day of the third tertile of SSB consumption was considered high as it is recommended by the Dietary Guidelines for Americans [30] to drink noncaloric water or beverages to the detriment of SSBs. An average of 6.5% in the third tertile of the SSB’s energy contribution was also considered high, and the total intake of added sugars can be higher because the sugars in other beverages and foods were not considered. In addition, according to the Food Guide for the Brazilian Population, the intake of sugars added to foods must be limited to 10% of the total daily calorie intake [31]. The World Health Organization also recommends limiting sugar intake to 5% of the total calorie intake [32].

A higher frequency of SSB consumption was associated with lower MMI in both sexes, which is in accordance with the findings of Hao et al. [6]. This study found that the consumption of sweetened beverages was associated with a 0.12 kg/m^2^ reduction in muscle mass in adolescents aged 14–18 years [6]. However, the SSBs considered in the analyses were not specified and the consumption of SSBs in the number of servings per day was considered.

The association between SSB consumption and reduced muscle mass can be explained by the fact that sugars in these beverages can interfere with muscle cell autophagy [7], telomere shortening, cell senescence, and apoptosis [8]. These findings may be correlated to muscle cell functionality and reduced muscle mass in male adolescents in the present study, evaluated both by the frequency of daily consumption and by the energy contribution of beverages.

In women, the association between the frequency of SSB intake and MMI was not observed after adjusting for carbohydrate, lipid, and protein intake. This phenomenon may be attributed to the fact that girls have less muscle mass [33] and the requirement for these macronutrients, particularly protein, has been supplemented by their diet because they are smaller in this sex. Moreover, although boys have a higher MMI, the intake of macronutrients may still be insufficient, thereby supporting the notion that the muscle mass is reduced with the consumption of sugars in SSBs.

In this study, an association was observed between the higher energy contribution from SSBs and lower MMI in male adolescents. However, although the energy contribution of SSBs in the third tertile was high, this association was not observed in females. However, a subsequent reduction in MMI among female adolescents may occur if they continue to consume high amounts of SSBs. Meanwhile, girls may have high SSB intake or eat other foods that have a minimal impact on total calories in the diet but may provide other nutrients that have not been investigated in this work and that may preserve muscle mass. Reverse causality may be another explanation, which is exemplified by the fact that adolescents with high body fat percentage and, consequently, reduced muscle mass had decreased SSB consumption.

Another interesting finding of this study is that the associations between reduced muscle mass and the highest frequency and energy contribution tertiary SSB in male adolescents were independent of the confounders studied but were mediated by sugars in the SSB.

An experimental study about the chronic consumption of sugars added to beverages and their effect on skeletal muscle has found a higher body weight, increased fat in the muscles, elevated serum glucose and insulin levels, and increased levels of triglyceride and interleukin-6 in the muscles after 1 month of ingestion of these sugars. An increased level of interleukin-6 was indicative of metabolic stress, which contributed to the activation of autophagy in muscle cells above normal physiological levels [7].

Some experimental studies have supported the findings showing that muscle fat is increased with the ingestion of sugars in SSBs [34,35]. The increase in the number and size of lipid droplets in muscle cells may contribute to muscle mass reduction as there is an adaptive mechanism with the simultaneous activation of lipolysis and greater autophagy in these cells [36].

Increased body fat accumulation also leads to impairment of biogenesis and mitochondrial function. Thus, dysfunctional mitochondria can generate a higher production of reactive oxygen species, which contribute to increased inflammation with a blockade of the mammalian target of rapamycin (mTOR) [37]. An experimental study found that a higher-sugar diet reduces mitochondrial function and muscle cell functionality [38]. These factors may explain the reduction in muscle mass observed in the results of the present study.

Another assumption is that a higher consumption of sugars contained in SSBs is correlated to a shortening of telomeres, which promote stability and protect DNA from damage. The length of telomeres naturally shortens with each cell cycle [8]. However, if exposed to oxidative stress and inflammation, the size of telomeres can be significantly reduced, and the cell may lose its ability to divide, leading to malfunction, aging, and cell death [39].

The frequency of daily consumption compared to energy contribution has physiological effects that contribute to a greater loss of muscle mass. When the daily consumption of SSBs is more often, the occurrence of hyperglycaemia and hyperinsulinemia and their physiological effects correlated to inflammation increases [40]. These situations lead to insulin resistance in muscle cells and lower muscle glycogen synthesis [41]. In relation to this, in the study by Petersen et al. [42], individuals with insulin resistance in the skeletal muscle were shown to have a 60% reduction in muscle glycogen synthesis after consuming a high-sugar diet, which may cause the reduced MMI in both male and female participants who more often consumed SSBs.

When assessing the effect size of the associations found in the present study, its significance was minimal but important because the population assessed was young and was in the phase of life in which reduction in muscle mass is not expected. Thus, more significant associations between SSBs and MMI may be observed later if these adolescents continue to consume SSBs in large quantities. The concern is that muscle mass is inversely correlated to overall mortality regardless of body weight, and its reduction compromises immune competence and proper functioning of the body [43].

The strengths of this study include the inclusion of a relatively large sample size, which involved population-based data, use of validated and standardized instruments, and DXA as an accurate method for estimating muscle mass. The use of MMI as an indicator of muscle mass was also a strength of the study, as it allowed the elimination of differences correlated to the height of adolescents. In addition, the use of two indicators to assess SSB consumption is highlighted, one representing the frequency of SSB intake and the other correlated to the percentage of calories in the diet.

However, the present study also had some limitations. Although the data used in this study were from a cohort initiated at birth, this was a cross-sectional study. Thus, causality between the consumption of SSBs and MMI is challenging to assess. In addition, the use of self-reported SSB measures might have led to information bias.

The intake of sugars in these drinks may be the mechanism underlying the association found in men.

## 5. Conclusions

An association was observed between reduced MMI and a higher frequency of SSB consumption in both male and female adolescents and between reduced MMI and a higher energy contribution of SSBs in male participants alone. Moreover, the intake of sugars in these beverages could be the mechanism underlying the association found in men, while in women, macronutrient intake could be contributed to the associations found. The consumption of SSBs was high in some adolescents, and it is considered a risk factor of reduced MMI and may be an important target in the development of strategies to reduce or eliminate the intake of these beverages to prevent the progressive loss of muscle mass with increasing age.

## Figures and Tables

**Figure 1 nutrients-14-04917-f001:**
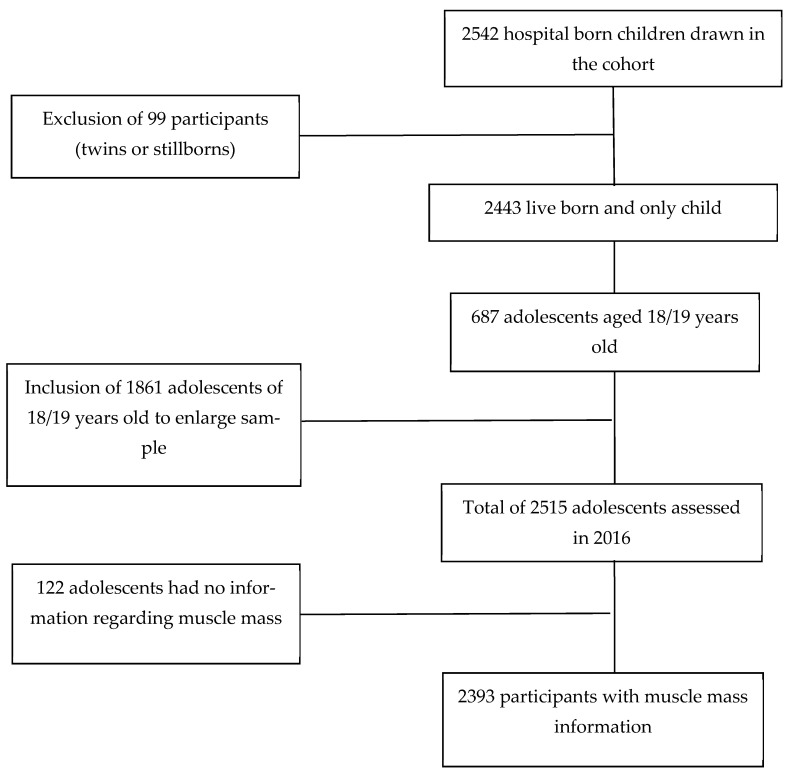
Flowchart of this study about the association between sugar-sweetened beverages and muscle mass index, Sao Luis, Brazil, 2016.

**Table 1 nutrients-14-04917-t001:** Sociodemographic characteristics and lifestyle habits of adolescents according to sex. Sao Luis, Brazil, 2016.

Variables	Male		Female		Total	
	%	n	%	n	%	n
Age (years) (*n* = 2393) *						
18	70.9	804	66.1	833	68.4	1637
19	29.1	329	33.9	427	31.6	756
Skin color (*n* = 2379)						
White	18.2	205	20.8	261	19.6	466
Black	16.2	183	16.5	206	16.3	389
Brown	65.6	739	62.7	785	64.1	1524
Schooling (*n* = 2393) *						
Elementary	1.2	14	0.6	7	0.9	21
High school	69.6	789	63.7	803	66.5	1592
Technical/vocational course	4.6	52	5.9	74	5.3	126
College	24.6	278	29.8	376	27.3	654
Economic class (*n* = 2120) *						
A	9.1	90	6.9	78	7.9	168
B	51.5	510	46.1	520	48.6	1030
C	38.0	377	45.2	511	41.9	888
D–E	1.4	14	1.8	20	1.6	34
Physical activity (*n* = 2386) *						
Insufficiently active	58.8	664	19.2	242	38.0	906
Physically active	41.2	465	80.8	1015	62.0	1480
Sleep duration (*n* = 2391)						
Less than 8 h	55.5	628	55.9	704	55.7	1332
8 h or more	44.5	503	44.1	556	44.3	1059
Alcohol consumption (*n* = 2381) *						
No	52.9	596	62.7	787	58.1	1383
Yes	47.1	530	37.3	468	41.9	998
Smoking (*n* = 2393) *						
No	95.1	1077	97.9	1233	96.5	2310
Yes	4.9	56	2.1	27	3.5	83
Total	100	1133	100	1260	100	2393

Legend: * *p*-value < 0.01.

**Table 2 nutrients-14-04917-t002:** Consumption of sugar-sweetened beverages, macronutrients, and sugar in beverages and muscle mass index in adolescents according to sex. Sao Luis, Brazil, 2016.

Variables	Male		Female		Total	
	Mean	SD	Mean	SD	Mean	SD
MMI (kg/m^2^) *	16.3	2.0	13.3	1.6	14.7	2.4
Frequency of Daily SSB Consumption (times/day)						
Chocolates drinks (times/day)	0.3	0.5	0.4	0.6	0.4	0.6
Soft drinks (times/day)	0.4	0.6	0.4	0.5	0.4	0.6
Industrialized Juices (times/day)	0.2	0.5	0.2	0.5	0.2	0.5
SSB (times/day)	0.9	1.0	0.9	1.0	0.9	1.0
SSB tertile 1 (0–0.3 times/day)	0.2	0.1	0.2	0.1	0.2	0.1
SSB tertile 2 (0.4–1.0 times/day)	0.6	0.2	0.7	0.2	0.6	0.2
SSB tertile 3 (1.1–10.1 times/day)	2.1	1.2	2.0	1.0	2.0	1.1
Energy contribution of SSB (% total energy intake)						
Chocolates drinks (%)	1.9	2.5	2.0	2.2	1.9	2.2
Soft drinks (%)	1.4	1.9	1.4	2.0	1.4	1.9
Industrialized juices (%)	1.2	2.2	1.2	2.5	1.2	2.4
SSB (%)	2.9	3.3	2.7	3.5	2.8	3.4
SSB tertile 1 (0–0.9%)	0.3	0.3	0.3	0.3	0.3	0.3
SSB tertile 2 (1.0–2.9%)	1.8	0.6	1.8	0.6	1.8	0.6
SSB tertile 3 (3.0–28.4%)	6.2	3.6	6.7	4.0	6.5	3.8
Energy intake (kcal/day) *	3217.6	1613.2	2844.7	1489.2	3021.2	1559.9
Carbohydrate consumption (%) *	63.2	6.3	61.7	6.0	62.4	6.2
Lipid consumption (%) *	22.7	5.0	24.2	5.2	23.5	5.2
Protein consumption (%)	14.0	3.1	14.0	3.1	14.0	3.1
Carbohydrate consumption (g/day) *	504.9	244.5	437.6	225.7	469.5	237.1
Lipid consumption (g/day) *	83.6	53.2	78.9	52.2	81.1	52.7
Protein consumption (g/kg)	1.7	1.0	1.7	1.0	1.7	1.0
Sugars in SSB (g/day)	32.8	39.5	30.0	38.6	31.4	39.1

Legend: SD: standard deviation. MMI: muscle mass index. SSB: sugar-sweetened beverage. * *p*-value < 0.05.

**Table 3 nutrients-14-04917-t003:** Association between the frequency of daily consumption of sugar-sweetened beverages and muscle mass index in adolescents according to sex. Sao Luis, Brazil, 2016.

		MMI						
		Male				Female		
SSB	β ^a^	95% CI	*p* Value	Stand. β ^b^	β ^a^	95% CI	*p* Value	Stand. β ^b^
Model 1								
1st tertile	1.00				1.00			
2nd tertile	−0.22	−0.50; 0.07	0.134	−0.052	0.01	−0.23; 0.22	0.950	−0.002
3rd tertile	−0.31	−0.60; −0.01	0.042	−0.070	−0.24	−0.45; −0.02	0.032	−0.068
Model 2								
1st tertile	1.00				1.00			
2nd tertile	−0.31	−0.62; 0.00	0.051	−0.072	0.09	−0.23; 0.25	0.937	0.003
3rd tertile	−0.44	−0.76; −0.11	0.008	−0.100	−0.27	−0.50; −0.03	0.026	−0.075
Model 3								
1st tertile	1.00				1.00			
2nd tertile	−0.32	−0.63; −0.01	0.043	−0.073	0.01	−0.23; 0.25	0.938	0.005
3rd tertile	−0.46	−0.79; −0.14	0.005	−0.110	−0.27	−0.50; −0.03	0.026	−0.076
Model 4								
1st tertile	1.00				1.00			
2nd tertile	−0.33	−0.62; −0.03	0.031	−0.073	0.01	−0.21; 0.24	0.904	0.008
3rd tertile	−0.47	−0.81; −0.13	0.007	−0.105	−0.19	−0.42; 0.04	0.109	−0.051
Model 5								
1st tertile	1.00				1.00			
2nd tertile	−0.27	−0.58; 0.03	0.078	−0.062	0.06	−0.17; 0.29	0.609	0.020
3rd tertile	−0.26	−0.69; 0.17	0.241	−0.060	−0.01	−0.31; 0.28	0.944	−0.003

Legend: ^a^ β: coefficient of linear regression. ^b^ β stand.: standardized coefficient to evaluate effect size. MMI: muscle mass index. SSB: sugar-sweetened beverage (soft drinks, industrialized juices, and chocolate drinks). Model 1: unadjusted analysis; Model 2: adjusted for age, skin color, education, and socioeconomic status; Model 3: adjusted for model 2 variables as well as physical activity, smoking status, alcohol consumption, and sleep duration; Model 4: adjusted for model 3 variables as well as carbohydrate, lipid, and protein intake; Model 5: adjusted for model 4 variables as well as intake of sugar in SSBs.

**Table 4 nutrients-14-04917-t004:** Association between the energy contribution of sugar-sweetened beverages and muscle mass index in adolescents according to sex. Sao Luis, MA, 2016.

		MMI						
		Male				Female		
SSB	β ^a^	95% CI	*p* Value	Stand. Β ^b^	β ^a^	95% CI	*p* Value	Stand. Β ^b^
Model 1								
1st tertile	1.00				1.00			
2nd tertile	−0.26	−0.55; 0.03	0.085	−0.062	0.08	−0.13; 0.30	0.454	0.023
3rd tertile	−0.34	−0.64; −0.04	0.024	−0.080	−0.05	−0.27; 0.16	0.632	0.015
Model 2								
1st tertile	1.00				1.00			
2nd tertile	−0.33	−0.65; −0.01	0.043	−0.078	0.08	−0.16; 0.32	0.511	0.020
3rd tertile	−0.43	−0.76; −0.10	0.010	−0.100	−0.07	−0.31; 0.16	0.541	−0.021
Model 3								
1st tertile	1.00				1.00			
2nd tertile	−0.28	−0.60; 0.03	0.079	−0.063	0.15	−0.09; 0.38	0.288	0.042
3rd tertile	−0.42	−0.74; −0.09	0.012	−0.090	−0.02	−0.26; 0.21	0.862	−0.002
Model 4								
1st tertile	1.00				1.00			
2nd tertile	−0.39	−0.69; −0.08	0.013	−0.086	0.07	−0.15; 0.29	0.536	0.023
3rd tertile	−0.52	−0.84; −0.21	0.001	−0.111	−0.04	−0.26; 0.18	0.712	−0.006
Model 5								
1st tertile	1.00				1.00			
2nd tertile	−0.36	−0.66; −0.05	0.024	−0.078	0.11	−0.11; 0.33	0.319	0.034
3rd tertile	−0.38	−0.75; 0.01	0.081	−0.076	0.15	−0.11; 0.41	0.265	0.045

Legend: ^a^ β: coefficient of linear regression. ^b^ β stand.: standardized coefficient to evaluate effect size. MMI: muscle mass index. SSB: sugar-sweetened beverage (soft drinks, industrialized juices, and chocolate drinks). Model 1: unadjusted analysis; Model 2: adjusted for age, skin color, education, and socioeconomic status; Model 3: adjusted for model 2 variables as well as physical activity, smoking status, alcohol consumption, and sleep duration; Model 4: adjusted for model 3 variables as well as carbohydrate, lipid, and protein intake; Model 5: adjusted for model 4 variables as well as intake of sugar in SSBs.

## Data Availability

The data that support the findings of this study are available by e-mail: rosangela.flb@ufma.br, but restrictions apply to the availability of these data, which were used under license for the current study, and so are not publicly available. Data are, however, available from the authors upon reasonable request and with permission from Rosangela Fernandes Lucena Batista.

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
