# Peer review of "The Frequency of Daily Consumption of Sugar-Sweetened Beverages Is Associated with Reduced Muscle Mass Index in Adolescents"

_nutrients, 2022, doi:10.3390/nu14224917_

Round 1

Reviewer 1 Report

The manuscript by Barbosa Martins Braganca et al. describes a cross-sectional study to find associations between the consumption of sugar-sweetened beverages and muscle mass in 18-19 years old adolescents.

In general, the study is well-designed, and the total number of 2393 subjects provided a reasonable statistical power of ~77%. The topic is suitable for the journal “Nutrients” and of interest for public health. The manuscript per se is understandable and well structured. However, I do find a tendency to overinterpret the results, which should be improved before considering publication. In addition, there are several language issues that require a careful correction by a native speaker.

My specific comments and concerns are provided below

·       My main concern is the tendency to generalize the specific results found in the discussion section and the title of the manuscript. For example, the title just mentions “the consumption of SSB is associated with reduced muscle mass index in adolescents”, however, there were differences described in the frequency of consumption and the energy contribution. Further, the statistical model (and with that confounding factors) as well as the sex had an impact on the results. Thus, this needs to be specified. This is also true for the discussion- the model and the involved confounding factors are not mentioned here.

·       Figure 1: one of the text boxes does not contain English language, but Portuguese (?), further language issues need to be corrected (e.g. l.93 “was showed in the computer”)

·       Figure 1: the inclusion criteria for the further 1861 test persons remains unclear and should be added in the figure

·       The methods section should be split into different sub-sections (e.g. study design, cohort, statistical analysis….)

·       Table 3 & 4. The differences in the different models should be provided in the table legend, not only in the main text

·       Just a small comment: Acknowledgement: All authors are grateful to all teenagers, their fathers, and their mothers à I assume that the authors are grateful to the participants of the study, not all teenagers etc. in general

Author Response

Response letter

October 26, 2022

Article: CONSUMPTION OF SUGAR-SWEETENED BEVERAGES IS ASSOCIATED WITH REDUCED MUSCLE MASS INDEX IN ADOLESCENTS

Dear Reviewer,

We would like to thank you for reviewing and commenting on our article. Suggested changes are answered separately.

The following is a detailed description of how we addressed reviewers’ comments and suggestions.

In general, the study is well-designed, and the total number of 2393 subjects provided a reasonable statistical power of ~77%. The topic is suitable for the journal “Nutrients” and of interest for public health. The manuscript per se is understandable and well structured. However, I do find a tendency to overinterpret the results, which should be improved before considering publication. In addition, there are several language issues that require a careful correction by a native speaker. My specific comments and concerns are provided below:

My main concern is the tendency to generalize the specific results found in the discussion section and the title of the manuscript. For example, the title just mentions “the consumption of SSB is associated with reduced muscle mass index in adolescents”, however, there were differences described in the frequency of consumption and the energy contribution. Further, the statistical model (and with that confounding factors) as well as the sex had an impact on the results. Thus, this needs to be specified. This is also true for the discussion- the model and the involved confounding factors are not mentioned here.

We appreciate the reviewer’s comment. We changed it as requested (lines 2, 33-35, 280-281). The models and confounders involved were specified in the results (lines 194-197) and in the discussion (lines 256-260, 270-271, 289-290, 299-302*).

Figure 1: one of the text boxes does not contain English language, but Portuguese (?), further language issues need to be corrected (e.g. l.93 “was showed in the computer”)

We appreciate the reviewer’s comment. We changed it as requested (line 106). As recommended by the reviewer we hired a native speaker to check all the language issues. We request a new period to make these language adjustments.

Figure 1: the inclusion criteria for the further 1861 test persons remains unclear and should be added in the figure

We appreciate the reviewer’s comment. We changed it as requested (line 80-81).

The methods section should be split into different sub-sections (e.g. study design, cohort, statistical analysis….)

We appreciate the reviewer’s comment. We changed it as requested.

Table 3 & 4. The differences in the different models should be provided in the table legend, not only in the main text

We appreciate the reviewer’s comment. We changed it as requested.

Just a small comment: Acknowledgement: All authors are grateful to all teenagers, their fathers, and their mothers à I assume that the authors are grateful to the participants of the study, not all teenagers etc. in general.

We appreciate the reviewer’s comment. We changed it as requested (line 370).

Again, we welcome reviewers’ suggestions. We believe that they contributed to improving the article, which we would like to see published in the Nutrients.

Yours sincerely,

The authors.

Reviewer 2 Report

The authors should be commended for tackling an important topic and completing a lot of work.  Combining participants from multiple sources is ok, given the cross-sectional design. Major concerns, however, are noted below.  

If SSBs are the primary exposure/independent variable, why would it be necessary to control for sugar in the SSBs?  If there is a logic to that decision, it is not explained.

Why are SSB’s reported in times/day instead of something more meaningful like servings/day or grams/day?

It is unclear what is meant in table 2 by energy contribution of SSB (%).  If it is truly a mean of only 2.9% of energy coming from SSBs for males and 2.7% for females (as shown in table 2), it seems insignificant. 

Causal language in the discussion to explain the results is inappropriate for this cross-sectional study (for example, “the intake of sugars in these beverages is the mechanism underlying the association found in men”).

Author Response

Response letter

October 26, 2022

Article: CONSUMPTION OF SUGAR-SWEETENED BEVERAGES IS ASSOCIATED WITH REDUCED MUSCLE MASS INDEX IN ADOLESCENTS

Dear Reviewer,

We would like to thank you for reviewing and commenting on our article. Suggested changes are answered separately.

The following is a detailed description of how we addressed reviewers’ comments and suggestions.

If SSBs are the primary exposure/independent variable, why would it be necessary to control for sugar in the SSBs?  If there is a logic to that decision, it is not explained.

We appreciate the reviewer’s comment. We changed it as requested (line 148-150).

Why are SSB’s reported in times/day instead of something more meaningful like servings/day or grams/day?

We appreciate the reviewer’s comment. We've decided to study the frequency of daily consumption of SSBs due to the effects they have on the human body when consumed on a recurring basis throughout the day. These explanations are elucidated in the discussion (line 326-334).

It is unclear what is meant in table 2 by energy contribution of SSB (%).  If it is truly a mean of only 2.9% of energy coming from SSBs for males and 2.7% for females (as shown in table 2), it seems insignificant. 

We appreciate the reviewer’s comment. Thus, we can observe that the values ​​were very close, with no statistical differences in consumption between the sexes.

Causal language in the discussion to explain the results is inappropriate for this cross-sectional study (for example, “the intake of sugars in these beverages is the mechanism underlying the association found in men”).

We appreciate the reviewer’s comment. We changed it as requested.

Again, we welcome reviewers’ suggestions. We believe that they contributed to improving the article, which we would like to see published in the Nutrients.

Yours sincerely,

The authors.

Round 2

Reviewer 2 Report

I am satisfied with the changes.

Author Response

Dear Reviewer,

We would like to thank you for reviewing and commenting on our article. Suggested changes are answered separately.

The following is a detailed description of how we addressed reviewers’ comments and suggestions.

This is a nice piece of research and the authors have largely addressed the comments from reviewers. However, the manuscript still needs significant English language editing. In addition, a few minor suggestions:

- Please define what "Industrialized juices" are, preferably with examples

We appreciate the reviewer’s comment. We changed it as requested (lines 115-116).

- For clarity, rather than "chocolates", use "chocolate drinks" in text and tables

We appreciate the reviewer’s comment. We changed it as requested.

- Table 2: Clarify "Energy contribution" - is this % total energy intake?

We appreciate the reviewer’s comment. We changed it as requested.

- Table 2: Add total reported energy intake (Kcal or MJ / day) and preferably also % energy (not just grams/day) of carbohydrate, lipid and protein.

We appreciate the reviewer’s comment. We changed it as requested.

Again, we welcome reviewers’ suggestions. We believe that they contributed to improving the article, which we would like to see published in the Nutrients.

Yours sincerely,

The authors.
